# KAN-ViT: A Visual-Tactile Fusion Learning Method for Grasping States Classification and Safe Force Inference

1st Chunfang Liu
*Department of Information Science and Technology*
*Beijing University of Technology*
Beijing, China
cfliu1985@bjut.edu.cn

2nd Jiawei Sun
*Department of Information Science and Technology*
*Beijing University of Technology*
Beijing, China
sunjiawei@emails.bjut.edu.cn

*Abstract*—To ensure stable and safe grasping during fine operations, it is required that the robot can accurately determine the grasping states and infer safe operating force, especially when grasping deformable objects. However, when grasping soft and light objects, the tactile signal feedback is very weak, making it difficult to classify the grasping states. Aiming to solve this problem, we introduce the optical flow information into tactile image feature processing to capture subtle dynamic variations in tactile data. A multimodal optical flow dataset, OFB-6 is constructed to support feature-level fusion of visual and tactile modalities. Further, an improved end-to-end transformer architecture is proposed by integrating visual and tactile data for grasping states classification and safe force prediction. Specifically, the k-NN attention mechanism is employed to enhance grasp states classification accuracy and then replacing multilayer perceptions(MLPs) with KAN network for reducing computational complexity and improving time efficiency.

*Index Terms*—Deformable Objects, Optical Flow, Grasping States Classification, Safe Force Inference.

## I. INTRODUCTION

Currently, fine robotic operations are mainly applied to repetitive tasks in structured environments, where there is little uncertainty or deformation of manipulated objects [1]. It is still a research hotspot and challenging problem for operating soft objects or deformable objects such as fruits, vegetables [2], and bread since the prediction of appropriate operating force is required. Insufficient force may lead to slippage, resulting in the object falling, while excessive force could damage the object. From grasping states feedback to estimating safe and stable operating force is indispensable in these precision robotic tasks. Unlike common tactile sensors, the Gelsight visual-tactile sensor, with its internal camera, captures high-resolution images of contact geometry. This study combines data from the Gelsight sensor [3] and the external D435 camera to classify contact states of robotic fingertips and predict the reasonable contact force(see Fig. 1).

Our contributions are summarized as follows:
(1) For soft and lightweight objects, we introduce optical flow

The work was jointly supported by National Science Foundation of China (62273012), Scientific Research Project of Beijing Educational Committee (KM202110005023) and National Science Foundation of China (62373016).

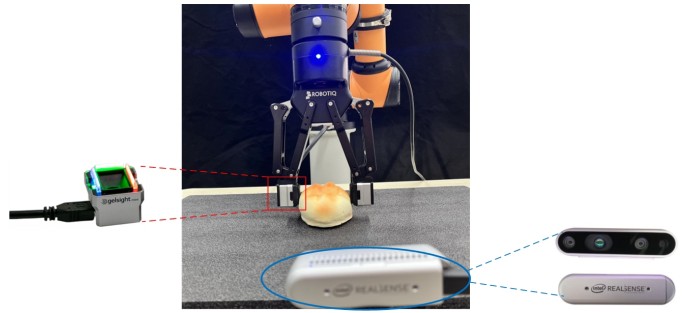

Fig. 1. The overall visual and tactile sensing system. Robot Setup: The AUBO-i5 robot is equipped with a gripper, where both fingers are integrated with Gelsight sensors. A Realsense D435 camera is positioned directly in front of the experimental platform.

information into tactile images to guide the model's learning and make a bread optical flow dataset, OFB-6.
(2) An end-to-end visual-tactile fusion model is proposed for grasping states classification and safe force inference. The model integrates the tactile and visual images by a transformer network, which is improved by incorporating the k-NN attention mechanism for enhancing the accuracy of grasping states classification.
(3) We further replace the Multi-Layer Perceptron (MLP) with the KAN neural network to improve the time efficiency of the model and the success rate of the safe force inference.

## II. RELATED WORK

### A. Grasping States Classification

Grasping states are typically classified into stable and unstable, with unstable states indicating imminent sliding. Recent studies[4][5][6][7][8][9][10] have employed various tactile sensing and machine learning techniques, including deep learning and visual-tactile sensors, to detect and prevent slip in robotic grasping, but most studies focused on rigid objects, which do not require precise force control.

### B. Visual-Tactile Robotic Learning

Human grasping involves both tactile sensation and visual observation to judge object slip states, with tac-

tile perception being the primary factor. Recent studies[11][12][13][14][15][16][17][18] have focused on integrating tactile and visual sensing through various deep learning and multimodal frameworks to enhance slip detection and grasp outcome prediction in robotic manipulation.

These studies highlight the advantages of visual-tactile learning, though they primarily focus on offline grasp states classification and lack exploration of online safe force adjustments. This paper aims to estimate safe grasping force for deformable objects, improving grasp stability and success rate. Yan et al. [19] proposed a transformer architecture for object slippage and safe grasping force detection. In contrast, this work focuses on softer, lighter objects, using a bread optical flow dataset (OFB-6) and incorporating optical flow information into tactile images. We propose the k-NNSformer architecture, replacing MLP with the KAN neural network for improved accuracy and efficiency. Additionally, our work correlates safe grasping force with actual force values through 3D force calibration.

## III. PROPOSED METHOD

In this section, we illustrate the details of the presented model for grasping states classification and safe force inference. In the subsection A, this paper innovatively integrates optical flow information into visual and tactile images as perceptual input to improve the effectiveness of feature extraction for grasping states classification. In the subsection B, we propose the KAN-ViT model by integrating k-NN attention mechanism and KAN network into the transformer model. Finally, the proposed model performs to estimate safe grasping force after two exploratory actions on the target object, namely pinching and lifting.

### A. Visual and tactile sensing

The sensing information in this work is gathered from three channels: visual images, tactile images, and optical flows extracted from tactile images.

*1) Visual images*: Fig. 1 shows the configuration of the visual and tactile sensing system. The blue oval box in Fig. 1 demonstrates the RealSense D435 visual camera mounted at the front of the robotic arm, which is used to capture the global state of the object.

*2) Tactile images*: A Gelsight mini visual-tactile sensor is installed at each end of the parallel gripper to capture subtle slip tendencies of the object. In fact, as shown in the red box of Fig. 1, only one of the tactile sensor is utilized to construct the minimum tactile perception system.

*3) Optical flows*: The optical flow can describe pixel motion in the image sequence by analyzing changes of pixel brightness on the image, which effectively reflects the dynamic motion of the object in contact.

### B. Proposed grasping states classification and safe force inference model

This subsection describes the proposed grasp states classification and safe force prediction framework in detail. The framework shown in Fig. 2 is primarily composed of five modules: KAN-ViT module, sensor fusion module, action fusion module, force threshold module and prediction module. (1) KAN-ViT Module: The image sequences from the sensor modalities are passed through the transformer model, producing vectors with size $D$. Therefore, for the two preset exploratory actions pinching and lifting, four vectors are generated: $v_{visual}^{pinch}$, $v_{tactile}^{pinch}$, $v_{visual}^{lift}$, and $v_{tactile}^{lift}$. (2) Sensor Fusion Module: After obtaining four vectors from the two modalities, the sensor fusion module concatenates each pair of vectors, resulting in $v^{pinch} = \left[ v_{visual}^{pinch}, v_{tactile}^{pinch} \right]$ and $v^{lift} = \left[ v_{visual}^{lift}, v_{tactile}^{lift} \right]$. (3) Action Fusion Module: The resulting vectors $v^{pinch}$ and $v^{lift}$ are fused into a single vector $v^{fused} = \left[ v_{visual}^{pinch}, v_{tactile}^{pinch}, v_{visual}^{lift}, v_{tactile}^{lift} \right] \in \mathbb{R}^{4 \times D}$. A linear transformation is then applied to project it into a low-dimensional space with an output size of $N$, as described by the linear transformation in Eq. 1.

$$Y^{fused} = v^{fused} \cdot W^{\mathrm{T}} + b \qquad (1)$$

Here, $Y^{fused} \in \mathbb{R}^{N \times 1}$ is the output vector, which represents a fused physical feature embedding. $W \in \mathbb{R}^{N \times 4D}$ is the weight matrix, a learnable parameter in the framework used to perform the linear mapping, and $b \in \mathbb{R}^{N \times 1}$ is another learnable bias, serving as the offset for the output.

(4) Force Threshold Module: GelSight is a vision-based tactile sensor and lacks the capability to directly estimate the grasping force. To address this issue, we conducted a 3D force calibration experiment to build the mapping relationship between the gel indentation depth and the corresponding normal force.

(5) Prediction Module: This module is mainly utilized for predicting suitable grasp force. The Prediction Module predicts an optimal grasp force based on low-dimensional physical embeddings from two exploratory actions and a force threshold candidate. It samples various grasping force candidates and inputs each threshold into the model, which predicts if the grasp will be stable. If multiple thresholds predict stability, their average is chosen. This process helps the robot select a safe and stable grasping force, preventing object damage.

## IV. CONCLUSION

In conclusion, this paper presents the KAN-ViT model, an effective visual-tactile fusion learning approach for grasping states classification and safe force inference. The model introduces two innovations: 1) integrating optical flow into tactile images to enhance weak sensing from soft objects and creating a bread optical flow dataset, and 2) developing an end-to-end visual-tactile fusion model for grasping states classification and safe force inference.

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
