# OpenReview forum: "KAN-ViT: A Visual-Tactile Fusion Learning Method for Grasping States Classification and Safe Force Inference"
_IEEE.org/IROS/2025/Workshop/Tactile_Sensing — IROS 2025 Workshop Tactile Sensing Poster_

### Official Review · Reviewer_7fsk · 2025-09-16
**A tactile-visual sensor fusion model for safe grasp**

**Rating:** 6
**Confidence:** 4

**Review:**

The authors present a framework for predicting safe grasp forces by combining visual and tactile inputs. The proposed approach is novel, incorporating visual and tactile encoders, a sensor fusion module, and a prediction model.

However, the reviewer recommends including preliminary results to validate the model’s effectiveness and justify the design choices. It would also be valuable to demonstrate the framework’s applicability across different objects and under different exploratory actions to strengthen the work.

---

### Official Review · Reviewer_oVGC · 2025-09-19
**Visual-Tactile Fusion Transformer**

**Rating:** 6
**Confidence:** 4

**Review:**

The authors present a visual-tactile fusion transformer model that takes both the RGB visual frames and Gelsight tactile frames as input for predicting suitable grasp force.

One recommendation is that the authors can add more details about the KAN module, especially how it prevails over the traditional MLP.

---

### Official Review · Reviewer_Nbsa · 2025-09-23
**A method with potential awaiting validation**

**Rating:** 6
**Confidence:** 4

**Review:**

This paper proposes a visual-tactile fusion model based on a transformer architecture enhanced with k-NN attention and KAN networks. The work addresses a problem of safe force inference for deformable objects.
A recommendation for strengthening the paper would be to include preliminary experimental results or a quantitative comparison. Even a small-scale validation demonstrating the model's accuracy or efficiency compared to a baseline (e.g., a standard ViT) would help substantiate the claimed improvements and clarify the practical benefit of the proposed KAN and k-NN attention mechanisms.